# Simulation of the NMR response in the pseudogap regime of the cuprates

Xi Chen[1], J.P.F. LeBlanc[1,2] & Emanuel Gull[1]

The pseudogap in the cuprate high-temperature superconductors was discovered as a suppression of the Knight shift and spin relaxation time measured in nuclear magnetic resonance (NMR) experiments. However, theoretical understanding of this suppression in terms of the magnetic susceptiblility of correlated itinerant fermion systems was so far lacking. Here we study the temperature and doping evolution of these quantities on the two-dimensional Hubbard model using cluster dynamical mean field theory. We recover the suppression of the Knight shift and the linear-in-$T$ spin echo decay that increases with doping. The relaxation rate shows a marked increase as $T$ is lowered but no indication of a pseudogap on the Cu site, and a clear downturn on the O site, consistent with experimental results on single layer materials but different from double layer materials. The consistency of these results with experiment suggests that the pseudogap is well described by strong short-range correlation effects.

[1] Department of Physics, University of Michigan, Ann Arbor, Michigan 48109, USA. [2] Department of Physics and Physical Oceanography, The Memorial University of Newfoundland, St John's, Newfoundland and Labrador, Canada, A1B 3X9. Correspondence and requests for materials should be addressed to X.C. (email: xichenli@umich.edu).

The pseudogap in the cuprates was discovered as a reduction of the Knight shift ($K_s$) and spin relaxation time $T_1$ measured in nuclear magnetic resonance (NMR) experiments[1–5]. Subsequent experimental research[6,7] has resulted in its detection in a wide range of materials and experimental probes for dopings smaller than optimal doping and temperatures smaller than 300 K. In single particle experiments, the pseudogap shows as a clear suppression of the density of states near the antinodal, but not the nodal, point of the Brillouin zone and is well described by non-perturbative numerical simulations[8–15] of fermion model systems.

Unlike single-particle probes, NMR provides a direct measure of a two-particle quantity, the magnetic (spin) susceptibility. The complete theoretical understanding of the two-particle signals measured in NMR is difficult, requiring two components: a precise relation of the NMR signal to correlation functions[16–18] and the low-energy spin susceptibility, and a reliable calculation of the spin susceptibility itself. While the first aspect has been well understood, directly obtaining the spin susceptibility of a correlated system has proven to be a formidable task, and as a result theoretical calculations of the NMR response have been limited to analytic or semi-analytic methods[19,20], high temperature[21] or sign-problem-free attractive models[22]. While these calculations provide a qualitative understanding of NMR signals outside the strong correlation regime, they either do not contain a pseudogap in the single particle quantities or are obtained on an attractive model and therefore are difficult to relate to the pseudogap in cuprate NMR spectra which is expected to originate from a repulsive interaction.

Recent advances in the numerical simulation of interacting fermionic lattice models[23] have made simulation of susceptibilities possible. In particular, a combination of cluster dynamical mean field methods[24] with continuous-time[25] auxiliary field[26,27] impurity solver extensions to two-particle functions[28] now allow for the unbiased calculation of generalized susceptibilities[29]. These methods are controlled, in the sense that they become exact as cluster size is increased, and treat short range correlations within the cluster exactly, while longer ranged correlations are approximated in a mean field way.

In this paper we employ these new techniques to obtain the spin susceptibility of the Hubbard Hamiltonian, $H = \sum_{\mathbf{k},\sigma}(\epsilon_{\mathbf{k}} - \mu)c_{\mathbf{k}\sigma}^{\dagger}c_{\mathbf{k}\sigma} + U \sum_i n_{i\uparrow}n_{i\downarrow}$ (see methods), from which we can compute all aspects of NMR probes. Specifically we show calculations for NMR experiments such as: the Knight shift, the spin echo decay time ($T_{2G}$) and the relaxation rate ($T_1^{-1}$). We focus on the temperature and doping dependence of these quantities for which a large body of experimental NMR work exists. Our results show remarkable similarity (both in temperature and doping dependence) to the experimentally measured quantities, indicating that the single-orbital Hubbard model, away from half filling and with an interaction strength close to the bandwidth captures much of the two-particle physics observed in experiment. Further, we isolate the role of pseudogap physics in each NMR probe.

## Results

**Knight shift.** In Fig. 1 we present the simulated NMR Knight shift as a function of temperature $T/t$ for a several dopings $x$. For these parameters, the largest $T_c$ on the hole doped side is $T_c = 0.03t$ at $x = 0.09$. At large doping $x = 0.1453$ (triangle, solid blue line), the simulated Knight shift monotonically increases as $T$ is reduced. Doped cases show a maximum at a temperature $T_{K_s}^*$, indicated by filled symbols. As the density decreases from $x = 0.0841$ to $x = 0$, this $T_{K_s}^*$ gradually moves to higher $T$. At all

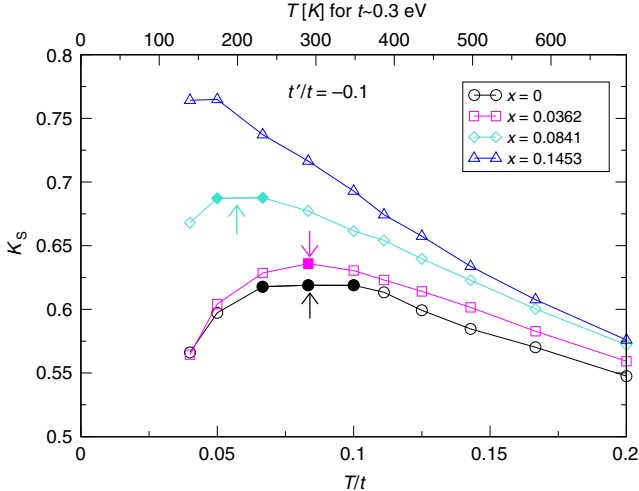

**Figure 1 | Knight shift.** Knight shift $K_S \propto \chi_m$ as a function of temperature $T/t$ (lower x axis) for a series of doping levels computed at $U = 6t$, $t' = -0.1t$ obtained from 8-site DCA. Filled symbols: the peak positions of the Knight shift. Arrows: onset of normal state pseudogap obtained by analytical continuation of the single particle spectral function at $K = (0, \pi)$. Upper x axis: $T/t$ in units of Kelvin assuming $t \sim 0.3$ eV.

temperatures studied, the overall magnitude of the Knight shift increases as doping is increased.

Several features in Fig. 1 described above are consistent with what is observed in NMR experiment on high $T_c$ cuprates, and we show side-by-side comparison to those data in the supplement. Firstly, in the underdoped regime the downturn of $\chi$ as $T$ is lowered is widely observed in $K_s$ at various nuclei sites, see, for example, Fig. 8 in ref. 4 on $YBa_2Cu_3O_{6.63}$ and Fig. 7 in ref. 30 on $YBa_2Cu_4O_8$; and similar data for $\chi(T)$ is found in squid magnetometry of $La_{2-x}Sr_xCuO_4$ (ref. 31), which has historically been interpreted as the onset of the pseudogap phase[4]. Secondly, the increasing Knight shift with increasing doping is observed in a wide range of compounds, including $La_{2-x}Sr_xCuO_4$ (refs 31,32), $YBa_2CuO_{7-x}$ and $YBa_2Cu_4O_8$ (ref. 18) and $Y_{1-x}Pr_xBa_2Cu_3O_7$ (ref. 18).

At high temperature, there is a distinct difference between the susceptibility measured in the bilayer material $YBa_2CuO_{6.63}$, which displays a broad maximum at 500 K and remains approximately constant up to 630 K (ref. 30), and that of the single layer material $La_{2-x}Sr_xCuO_4$ (refs 31,33), where measurements indicate a slowly decreasing Knight shift above $T^*$. This discrepancy may be caused by magnetic coupling of copper-oxygen planes in the bilayer materials. Our calculations, which are done on a purely two-dimensional system, are consistent with measurements performed on single layer materials.

**Extracting pseudogap energy scales.** The arrows in Fig. 1 indicate the onset temperature of the pseudogap in the single particle spectral function calculated by analytical continuation of the single particle Green's function[34] using the maximum entropy method[35,36]. From the temperature evolution of $A_{K=(\pi, 0)}(\omega)$, we define $T^*$ as the temperature at which a suppression of the density of states appears near zero frequency (see Supplementary Fig. 1). In agreement with refs 37,38, and as observed in a study of an attractive model[39], $T_{K_s}^*$ exhibits the same dependence on temperature and doping level as $T^*$, showing crossover temperatures identified with single-particle quantities (density of states) and two-particle quantities (Knight shift) to be the same. (While this work uses lattice susceptibility to calculate the

Knight shift, ref. 37 identifies $T^*$ based on cluster susceptibility, whose doping dependence is not consistent with NMR experiment.)

Figure 2 expands further upon the data in Fig. 1, including additional doping levels at $x = 0.0178$ and $x = 0.0585$ for temperatures above the superconducting $T_c$ and below $T^*_{K_s}$ as an Arrhenius plot. Once a gap has fully opened, the resulting curves become straight lines within uncertainties, allowing us to interpret our data as thermal excitations over a rigid gap and to extract an energy scale from the slopes using $\chi_m(T) = \chi_0 \exp(-\Delta_{pg(2p)}/T)$. The inset of Fig. 2 shows the comparison between the pseudogap energy determined by this method (open symbols) and the corresponding pseudogap energy extracted from the peak-to-peak distance of the single particle spectral function at the antinode (filled symbols). The two energy gaps are proportional as a function of doping. The distinct energy scales are however expected since $\Delta_{pg(2p)}$ averages over the Brillouin zone while $\Delta_{pg(1p)}$ only considers the antinodal momenta. As a result, their actual gap values in this case differ by a factor of 75, independent of doping. Similar comparisons for experimental data on YBa$_2$Cu$_4$O$_8$ yield values of $\Delta_{pg(1p)} \approx 150$ meV and $\Delta_{pg(2p)} = 7.75$ meV, a difference of a factor of 20 (refs 5,7,40–42). Potentially, a quantitative comparison of this ratio to experiment it might allow for a more precise determination of model parameters than considering single-particle properties alone.

**Spin echo decay time.** Figure 3 shows the spin echo decay time $T_{2G}$, a measure of indirect spin–spin coupling, calculated according to equation (5). Owing to the divergence of lattice susceptibility near $(\pi, \pi)$, we use the cluster susceptibility. This quantity shows a linear rise with temperature in the normal state and increases as doping is increased. The inset of Fig. 3 plots these data as $T_{2G}^{-1}$, the spin echo decay rate. $T_{2G}^{-1}$ becomes less temperature dependent as more charge carriers are added. Otherwise, and consistent with experiment, $T_{2G}$ is rather featureless in the normal state and shows no marked change upon entering the pseudogap region.

The linear increase of $T_{2G}$ depicted in Fig. 3 is similar to data obtained on YBa$_2$Cu$_4$O$_8$ in NMR experiments reported in Fig. 3

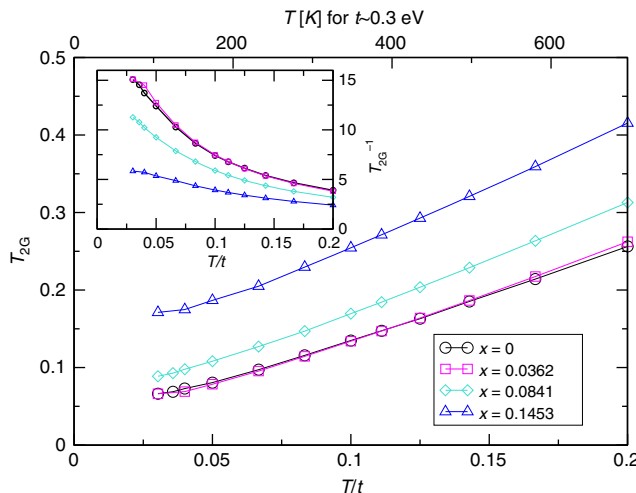

**Figure 3 | Spin echo decay rate.** Spin echo decay time $T_{2G}$ as a function of temperature for doping level ranging from $x = 0$ to $x = 0.145$, calculated at $U = 6t$, $t' = -0.1t$. Inset: spin echo decay rate $T_{2G}^{-1}$. Decay rates are in units of $(0.69/128)^{1/4}$ $^{63}\gamma_n$ where $^{63}\gamma_n$ is nuclear gyromagnetic ratio.

of ref. 43 and Fig. 3 of ref. 44, and NQR experiment (Fig. 4 of ref. 30). The change of magnitude of a factor of 4 from 100 to 700 K is comparable in this calculation and experiment. The increase of $T_{2G}^{-1}$ as charge carriers are added is similarly observed in YBa$_2$Cu$_3$O$_{7-x}$ experiment, see, for example, Fig. 8 of ref. 45 and Fig. 11 of ref. 18. We find no indication of a change of slope around $\sim 500$ K as discussed in Fig. 4 of ref. 30.

**Spin relaxation rate.** Figure 4 shows the simulated spin-lattice relaxation rate multiplied by the inverse temperature, $(T_1 T)^{-1}$, as a function of $T$ for three dopings (see equation (6)) with structure factors corresponding to copper and oxygen nuclei. All results are obtained at an interaction strength of $U = 6t$ using the cluster susceptibility. $(T_1 T)^{-1}$ for $^{63}Cu$ (solid line) rises rapidly when temperature is reduced. As doping is reduced, the value of $(T_1 T)^{-1}$ decreases, and no clear indication of the pseudogap onset temperature is visible. In contrast, $(T_1 T)^{-1}$ for $^{17}O$ (solid

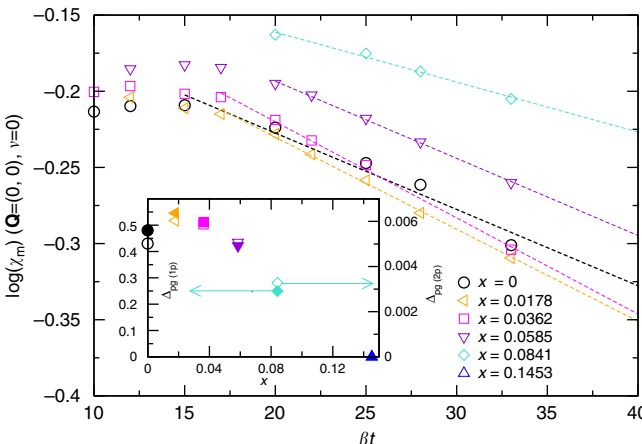

**Figure 2 | Extracting pseudogap energy scales.** Extraction of $\Delta_{pg(2p)}$ from Knight shift data via $\chi_m(T) = \chi_0 \exp(-\Delta_{pg(2p)}/T)$. Open symbols: data of Fig. 1 plotted as $log(\chi_m)$ versus $\beta$. Dashed lines: linear fits to the data in exponentially decaying regime. Inset: comparison between pseudogap energy extracted from the slope of Arrhenius plot (open symbols, right $y$ axis) and from the single particle spectral function at $\mathbf{K} = (0, \pi)$ (filled symbols, left $y$ axis).

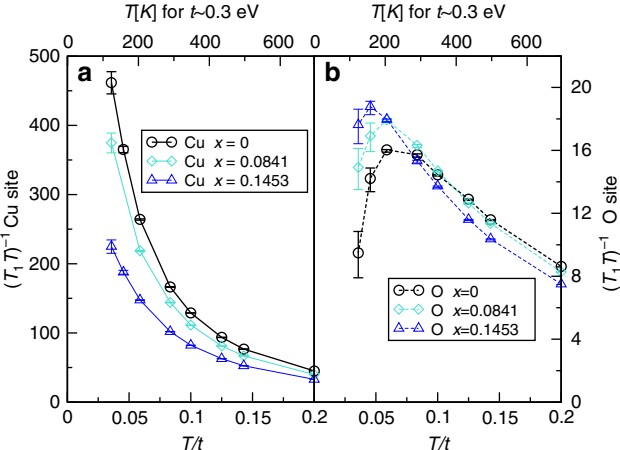

**Figure 4 | Spin-lattice decay rate.** $(T_1 T)^{-1}$ plotted as a function of temperature at $U = 6t$, $t' = -0.1t$, for $x = 0$ to $x = 0.145$, by 8-site DCA. (**a**) Solid lines: symmetry factors corresponding to $^{63}Cu$ site. (**b**) Dashed line: $^{17}O$ site (See Supplemental Material for explanation of uncertainties). Values in units of $(k_B/2\mu_B^2 \hbar^2)$[18].

line) has peaks at about the same temperatures as $T_{K_s}^*$. $(T_1T)^{-1}$ for both $^{63}Cu$ and $^{17}O$ become doping independent at even higher temperature (see Supplemental Material).

While reliable results for $T_1$ from other theoretical methods are absent in the pseudogap regime, our results can directly be compared to real-frequency RPA calculations for $T_1$ in the weak coupling regime[19]. These calculations are neither limited by the momentum resolution of dynamical cluster approximation (DCA), nor do they suffer from the limitations of analytic continuation. Therefore they provide a stringent check on the precision with which we can obtain relaxation rates. Our simulations show that $T_1^{-1}$ smoothly decreases towards zero as temperature is reduced, in good agreement with RPA for $U=2t$ (ref. 19), hinting at limitations of the random phase approximation in the intermediate coupling regime where deviations are apparent (see Supplemental Material).

The experimentally measured spin-lattice relaxation rates are strongly material dependent. One common feature found for the planar $Cu$ site in YBCO materials in the normal state is that $(T_1T)^{-1}$ increases slowly and linearly as $T$ decreases in a large range of temperature above $T^*$ (refs 4,40). As $T$ is lowered below $T^*$, it shows a decrease towards $T_c$. In contrast, experiments in LSCO materials show that $(T_1T)^{-1}$ for the planar $Cu$ site increases rapidly as temperature is decreased until $T_c$, with a larger rate as the doping level is decreased (see ref. 32, Fig. 4). $(T_1T)^{-1}$ data for planar $^{17}O$ in LSCO are proportional to the Knight shift in the range from 100 to 200 K (ref. 46). Doping-independent $(T_1)^{-1}$ is observed in NQR experiment on LSCO above 700 K (Fig. 2 in ref. 47), and NMR experiment on YBa$_2$(Cu$_{1-x}$Zn$_x$)$_4$O$_8$ above 150 K (Fig. 2 in ref. 48). A comparison of these two types of materials is made in ref. 45. Our result is consistent with the experimental result of LSCO and inconsistent with YBCO. We attribute this to the presence of interplanar spin couplings in the latter materials[49], whose existence is confirmed by neutron-scattering experiment[50], and surmise that more complicated bilayer models might be required to yield consistent result for the YBCO spin-lattice relaxation rates, also suggested from previous theoretical work[51].

**Temperature dependence of momentum-dependent susceptibility.** Expanding upon the previous section, we comment further on the distinct behaviour of the $^{63}Cu$ and $^{17}O$ signals in the calculated $(T_1T)^{-1}$ data of Fig. 4. Both calculations originate from the same spin susceptibility, and are distinguished only by the convolution with $^{63}F_{||}$ and $^{17}F_{||}$ structure factors, which are **q**-dependent functions.

In order to make these effects transparent, we present cuts in the **Q** = $(q_x, q_y)$ plane of the static spin susceptibility on the matsubara axis, $\chi(iv_0, q_x, q_y)$, in Fig. 5a. Plotted are 8-site DCA results, which therefore have only 4-distinct values at the $M$, $\Gamma$, $X$ points and at **Q** = $(\pi/2, \pi/2)$. As the temperature decreases, we see that the $X = (\pi, 0)$ point has no temperature dependence. There does exist temperature dependence at both **Q** = $(\pi/2, \pi/2)$ and $\Gamma$. However, the predominant effect with reduced temperature is the strong temperature dependence of the susceptibility near $M = (\pi, \pi)$, which shows a continual increase upon decreasing temperature. This is distinct behaviour from the $\Gamma$ point, which shows increasing behaviour until $T/t = T^* = 0.083$, after which it decreases (see inset of Fig. 5b. The contribution at $\Gamma = (0, 0)$ is precisely the NMR Knight shift (see Fig. 1).

The impact of the strong antiferromagnetic $(\pi, \pi)$ scattering vector is compounded by the influence of the Form factors $^{63}F_{||}$ and $^{17}F_{||}$. These are plotted in Fig. 5b. We see than for the Cu

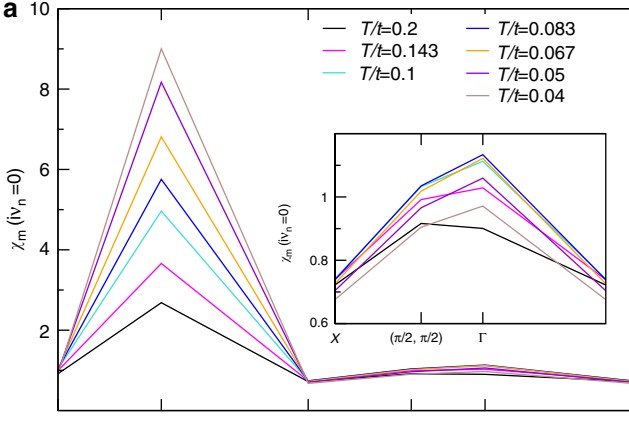

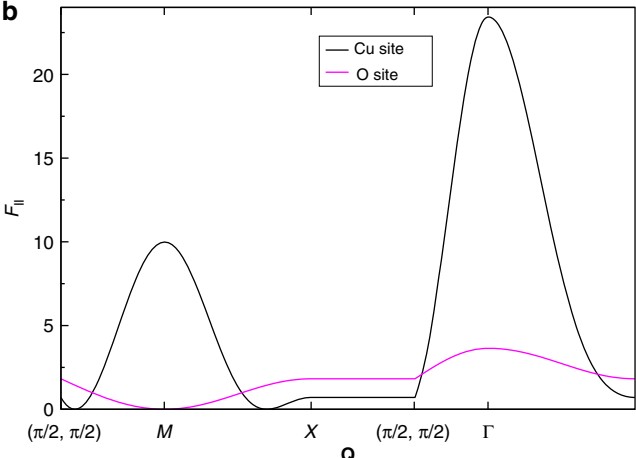

**Figure 5 | Structure factors.** (**a**) Zero frequency spin susceptibility at different momentum **Q** for various temperatures. Inset: The same data, zoomed in near **Q** = (0, 0). (**b**) The structure factor for copper ($^{63}F_{||}$) and oxygen ($^{17}F_{||}$) sites given by equation (7).

sites, the $^{63}F_{||}$ picks up both contributions of temperature dependence from $M = (\pi, \pi)$ and near $\Gamma = (0, 0)$. However, the signal is completely dominated by the monotonic increase with decreasing temperature of the susceptibility at the $M$-point. For oxygen sites, the $^{17}F_{||}$ suppresses the growth of the susceptibility near $(\pi, \pi)$, emphasizing instead the temperature dependence in the remaining regions around the $\Gamma$ point. It is for this reason that NMR probes show site selective pseudogap behaviour (observed on oxygen NMR and not in copper NMR).

**Discussion**

We have shown results for the doping and temperature evolution of the Knight shift, the relaxation time and the spin echo decay time in the pseudogap regime of the two-dimensional Hubbard model from state-of-the art numerical simulation. Our results were obtained using an eight-site DCA calculation that treats short ranged correlations exactly and approximates longer ranged correlations in a mean field way. In this work, our goal has not been explaining the original of the pseudogap, but instead elucidating what is actually being measured in NMR probes in the presence of a pseudogap. These calculations show trends in temperature and doping evolution that are in remarkable agreement with experiment on single layer compounds and deviate when compared to the relaxation rate of double layer compounds, indicating that both the relation of experimental

quantities to the generalized susceptibility and the calculation of the susceptibility in the pseudogap regime are well under control. Within these calculations, we demonstrate that the observation of a downturn in both Knight shift and $T_1^{-1}$ on oxygen sites originates from the same underlying effect, namely temperature dependence of the static susceptibility in the vicinity of $\mathbf{q} = (0, 0)$, and explain how such effects are washed out for $T_1^{-1}$ on copper sites.

Further, the agreement of the calculated two-particle quantities with NMR experiment and relation to single-particle features of the pseudogap ($T^*$ and $\Delta_{\mathrm{pg}}$) suggests that the salient aspects of the physics of the cuprate pseudogap are contained within the simple single-orbital Hubbard model[14,52]. Phenomena absent from this calculation, for example, stripes, charge ordering, multi-orbital effects[53,54] or nematic order, may occur in addition to the physics realized here but do not seem to be the primary cause of the pseudogap observed in the cuprates via NMR probes.

The marked difference between multi layer and single layer materials suggests that inter-layer correlations, absent in these calculations, have a strong effect on the relaxation time. Providing reliable calculations of such effects, along with providing a description of the full momentum and frequency dependence of the susceptibility measured in neutron spectroscopy[55], remains an interesting open challenge.

## Methods
### Susceptibilities.
We study the single orbital Hubbard model in two dimensions with nearest and next nearest hopping parameters in the normal state,

$$H = \sum_{\mathbf{k},\sigma} (\epsilon_{\mathbf{k}} - \mu) c_{\mathbf{k}\sigma}^{\dagger} c_{\mathbf{k}\sigma} + U \sum_i n_{i\uparrow} n_{i\downarrow}, \tag{1}$$

where $\mu$ is the chemical potential, $\mathbf{k}$ momentum, $i$ labels sites in real-space, $U$ the interaction and the dispersion is given by $\epsilon_{\mathbf{k}} = -2t[\cos(k_x) + \cos(k_y)] - 4t'\cos(k_x)\cos(k_y)$.

The generalized susceptibility $\chi$ (refs 56,57) is written in imaginary time in terms of the one-particle $G_{\sigma_1\sigma_2}(\mathbf{k}_1\tau_1, \mathbf{k}_2\tau_2) = \langle T_\tau(c_{\mathbf{k}_1\sigma}^{\dagger}(\tau_1) c_{\mathbf{k}_2\sigma}(\tau_2).\rangle$ and two-particle $G_{2,\sigma_1\sigma_2\sigma_3\sigma_4}(\mathbf{k}_1\tau_1, \mathbf{k}_2\tau_2, \mathbf{k}_3\tau_3, \mathbf{k}_4\tau_4) = \langle T_\tau(c_{\mathbf{k}_1\sigma}^{\dagger}(\tau_1) c_{\mathbf{k}_2\sigma}(\tau_2) c_{\mathbf{k}_3\sigma}^{\dagger}(\tau_3) c_{\mathbf{k}_4\sigma}(\tau_4).\rangle$ Green's functions as $\chi_{\sigma_1\sigma_2\sigma_3\sigma_4}(\mathbf{k}_1\tau_1, \mathbf{k}_2\tau_2, \mathbf{k}_3\tau_3, \mathbf{k}_4\tau_4) = G_{2,\sigma_1\sigma_2\sigma_3\sigma_4}(\mathbf{k}_1\tau_1, \mathbf{k}_2\tau_2, \mathbf{k}_3\tau_3, \mathbf{k}_4\tau_4) - G_{\sigma_1\sigma_2}(\mathbf{k}_1\tau_1, \mathbf{k}_2\tau_2) G_{\sigma_3\sigma_4}(\mathbf{k}_3\tau_3, \mathbf{k}_4\tau_4)$. Its Fourier transform is

$$\chi_{ph\sigma\sigma'}^{\omega\omega'\nu}(\mathbf{k}, \mathbf{k}', \mathbf{q}) = \int_0^\beta d\tau_1 d\tau_2 d\tau_3 e^{-i\omega\tau_1 + i(\omega+\nu)\tau_2 - i(\omega'+\nu)\tau_3} \times \chi_{\sigma\sigma'\sigma'}(\mathbf{k}\tau_1, (\mathbf{k}'+\mathbf{q})\tau_2, (\mathbf{k}+\mathbf{q})\tau_3, k'0) \tag{2}$$

where $\omega$ and $\omega'$ are fermionic Matsubara frequencies, $\nu$ is a bosonic Matsubara frequency, $\sigma$ and $\sigma'$ are $\uparrow$ or $\downarrow$ spin labels and $\mathbf{k}$, $\mathbf{k}'$ and $\mathbf{q}$ are initial, final and transfer momenta respectively. $ph$ denotes the Fourier transform convention[56]. The main object of interest, the spin susceptibility, is then defined as

$$\chi_m = \chi_{ph\uparrow\uparrow} - \chi_{ph\uparrow\downarrow}. \tag{3}$$

### Knight shift.
According to the Mila–Rice–Shastry model[16,17] for hyperfine coupling with itinerant $Cu^{2+}$ holes in high $T_c$ cuprates, the Knight shift $K_S$ measured in NMR experiment is proportional to the uniform spin susceptibility,

$$K_S \propto \chi_m(\mathbf{q} = (0, 0), \nu = 0). \tag{4}$$

### Spin echo decay rate.
For the $^{63}Cu$ nuclear spin echo decay rate $^{63}1/T_{2G}$ in paramagnetic state of high $T_c$ cuprates[58] showed that

$$^{63}T_{2G}^{-2} = \frac{0.69}{128\hbar^2} \left[ \frac{1}{N} \sum_{\mathbf{q}} {}^{63}F_{\mathrm{eff}}(\mathbf{q})^2 \chi_m'(\mathbf{q}, 0)^2 - \left( \frac{1}{N} \sum_{\mathbf{q}} {}^{63}F_{\mathrm{eff}}(\mathbf{q}) \chi_m'(\mathbf{q}, 0) \right)^2 \right], \tag{5}$$

where $\chi_m'(\mathbf{q}, \nu = 0)$ denotes the real part of the real-frequency dynamical spin susceptibility at momentum $\mathbf{q}$ and frequency $\nu = 0$ and $N$ is number of $\mathbf{q}$ points sampled in the first Brillouin zone. The prefactor 0.69 originates from the natural abundance of $^{63}Cu$ (ref. 59), and $^{63}F_{\mathrm{eff}}$ is defined in ref. 18 with hyperfine coupling constants $A$ and $B$ as $^{63}F_{\mathrm{eff}} = \{A_{\parallel} + 2B[\cos(q_x a) + \cos(q_y a)]\}^2$, $A_{\parallel} = -4B$. For simplicity we set $B \equiv 1$ and consider only proportionality. With this, both $K_S$ and

$T_{2G}$ can be calculated directly from a susceptibility on the Matsubara axis since $\chi(\mathbf{q}, \nu = 0) = \chi(\mathbf{q}, i\nu = 0)$.

### Spin-lattice relaxation rate.
The spin-lattice relaxation rate $1/T_1$ is related to the imaginary part of dynamical spin susceptibility on the real axis,

$$\frac{1}{T_1 T} \propto \lim_{\nu \to \infty} \sum_{\mathbf{q}} {}^{\alpha}F_{\parallel}(\mathbf{q}) \frac{\chi_m''(\mathbf{q}, \nu)}{\nu}. \tag{6}$$

where $^{\alpha}F_{\parallel}(\mathbf{q})$ differs for $^{63}Cu$ and $^{17}O$, as defined in ref. 18.

$$\begin{aligned} {}^{63}F_{\parallel} &= \left\{ A_{\perp} + 2B[\cos(q_x) + \cos(q_y)] \right\}^2 \\ {}^{17}F_{\parallel} &= 2C_{\parallel}^2 \left[ 1 + 0.5[\cos(q_x) + \cos(q_y)] \right] \\ A_{\perp} &= 0.84B, \quad C_{\parallel} = 0.91B. \end{aligned} \tag{7}$$

The calculation of $\chi_m''(\mathbf{q}, \nu)/\nu$ within a Matsubara formalism requires analytical continuation[35]. However, the quantity $S(\mathbf{q}, \tau)$, defined as the real-to-k-space Fourier transform of $S_z = n_{i\uparrow} - n_{i\downarrow}$, satisfies

$$\sum_{\mathbf{q}} {}^{\alpha}F_{\parallel}(\mathbf{q}) S\left(\mathbf{q}, \tau = \frac{1}{2T}\right) = \sum_{\mathbf{q}} {}^{\alpha}F_{\parallel}(\mathbf{q}) \int d\nu \frac{\chi_m''(\mathbf{q}, 0)}{\nu} \frac{\nu}{\sinh\frac{\nu}{2T}}$$

(see ref. 22 for details) such that in the limit $T \to 0$

$$\frac{1}{T_1} \overset{T \to 0}{\propto} \sum_{\mathbf{k}\mathbf{k}'\mathbf{q}\omega\omega'\nu} \frac{2}{\pi^2 N\beta^2} {}^{\alpha}F_{\parallel}(\mathbf{q}) \chi_m^{\omega\omega'\nu}(\mathbf{k}, \mathbf{k}', \mathbf{q}) e^{\frac{-i\nu}{2T}}. \tag{8}$$

### Numerical method.
The direct numerical solution of equations (4), (5) and (8) for the Hubbard Hamiltonian equation (1) is intractable. We therefore employ the DCA[24] which approximates the momentum dependence of the many-body self-energy and irreducible vertex functions by an approximated function that is constant on a set of $N_c$ 'patches' in momentum space[24,60]. The method is a non-perturbative short correlation length approximation and is controlled in the sense that as $N_c$ is increased it converges to the exact limit[23,27,61]. Throughout this paper we use $N_c = 8$, a compromise between accuracy and efficiency that has previously been shown to capture much of the single-particle[12,13,34,62] and two-particle[28,29] physics observed in experiment and shows a qualitatively correct phase diagram for the pseudogapped and superconducting phases[63–66]. The interaction strength $U = 6t$ is large enough to exhibit a clear pseudogap state but presumably slightly smaller than seen in experiment, having an optimal doping and pseudogap onset closer to half filling[29]. We use a next-nearest neighbour hopping of $t' = -0.1t$, and do not allow for long ranged ordered antiferromagnetic or superconducting states.

### Data availability.
The computer code and data that support the findings of this study are available from the corresponding author on request.

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

## Acknowledgements

We thank R. Walstedt and A.J. Millis for insightful and productive discussions on experimental data and on theory. We acknowledge the Simons Collaboration on the Many-Electron Problem for methods development and NSF grant 1606348 for application to NMR. Our simulations made use of the ALPS[67] library and were performed on XSEDE using TG-DMR130036.

## Author contributions

X.C. performed the simulation and prepared the figures. X.C., J.P.F.L. and E.G. implemented computer codes, analysed the data and wrote the manuscript.

## Additional information

**Competing interests:** The authors declare no financial competing interests.

**Publisher's note**: 

