## [Peer Review File · Nature Communications]

Reviewers' comments:

Reviewer #1 (Remarks to the Author):

The authors study the NMR Knight shift and spin relaxation rate in the pseudo gap regime of the copper oxides, via treating an effective two dimensional Hubbard model.

The relaxation rate does not show any pseudo gap feature for the Cu site, and show a down turn for the O site. The authors conclude that the one band Hubbard model with short range interaction is enough to describe consistently the physics of the copper oxides. The authors use a mean field approach (cluster dynamical mean field theory) and a model with nearest and next-nearest hoppings.

The paper is generally well written, but however I find the conclusions slightly misleading, and I think the authors should consider extending the discussion of their results and how they generally connect with the literature.

A few points that the author might integrate in the discussion:

- 1) The 2D Hubbard model is believed to yield a description of most copper oxides. How would the authors reconcile their results with the observation by polarized neutron of a local magnetic moment in the O-Cu-O in plane plaquette in the pseudogap phase of some of the copper oxides ? (see Ref [1] below)
- 2) minor point , but in formula (5) "N" isn't defined.
- 3) another minor point, it would be useful to have the units of T1 and T2.
- 4) I think it should be emphasized in the paper that only short range fluctuations are incorporated in the calculations. Extending the method beyond DMFT to GW would incorporate long range screening processes, which aren't accounted for by the present methodology.
- 5) I don't get the comment of the author that multi-orbital effects aren't important ?
- 6) Maybe it would be worth mentioning what is the pseudo-gap obtained for $U=6t$, and also maybe mentioning what is the charge gap of the undoped model. Those quantities will have been published somewhere else, but reminding them in this paper would be helpful.

If the authors can address these points, I'll be happy to recommend publication.

[1] <http://www.nature.com/nature/journal/v468/n7321/abs/nature09477.html>

Reviewer #2 (Remarks to the Author):

This is a very interesting paper that should be published. The authors provide strong evidence that the pseudogap is predominantly associated with short-range magnetic order, and that other effects, such as "stripes, multi-orbital effects or nematic order ... do not seem to be the primary cause of the pseudogap". [This presumably also goes for charge-density wave order, which cluster calculations have difficulty seeing.] The authors also provide a careful comparison to the experimental data.

I have only a few suggestions, to clarify points for the reader:

(1) lines 118-9: "Due to the divergence of lattice susceptibility near (π, π) , we use the cluster susceptibility." This confuses me -- the cluster calculation is supposed to get better as the size of the cluster increases. So how do we know that the magnetic order is really short-range, and not a long-range order that is being smeared out by the cluster averaging? Of course, the experiment says the order is only short range, but does the theory predict this? This should be carefully discussed.

(2) lines 123-4: " T_{2g} ... shows no marked change upon entering the pseudogap regime." In this instance, I don't see a clear comparison to experiment [unless that is what lines 130-1 are hinting.]

Housekeeping:

(1) there are 2 typos on line 207.

(2) The supplementary material has numerous typos, and some overlap with the main text, particularly Methods Section and figures. I believe some overlap is OK, if it is intended to maintain clarity for the reader. However, it would be good to go over this material once more, to make sure there are no unintended overlaps.

REVIEWERS' COMMENTS:

Reviewer #1 (Remarks to the Author):

The authors answered all my queries, I am happy to recommend publication as it stands.

Reviewer #2 (Remarks to the Author):

The authors have addressed my concerns, and I believe the manuscript is now acceptable for publication, providing a striking interpretation of NMR results in cuprates.

Reply to Referees: Simulation of NMR responses in the pseudogap regime of the cuprates

We hereby resubmit our revised version of the manuscript ‘Simulation of Knight shift, spin-echo and spin-relaxation rates in the pseudogap regime of the cuprates’. As requested we attach a complete checklist, a detailed response to the comments by the referees, and a version of the manuscript with highlighted changes. We believe that our revised version addresses all the comments raised by the referee, and hope that our paper can now be published.

RESPONSE TO REFEREE 1

Referee A remarks that ‘The paper is generally well written’ and that he ‘will be happy to recommend publication’ if his concerns are addressed. We thank the referee for this positive assessment and answer the six points he raised below in detail.

- The 2D Hubbard model is believed to yield a description of most copper oxides. How would the authors reconcile their results with the observation by polarized neutron of a local magnetic moment in the O-Cu-O in plane plaquette in the pseudogap phase of some of the copper oxides?

Our precise wording was: *The agreement of the calculated two-particle quantities with NMR experiment and relation to single-particle features of the pseudogap (T^* and Δ_{pg}) suggests that the salient aspects of the physics of the cuprate pseudogap are contained within the simple single-orbital Hubbard model^{1,2}. Phenomena absent from this calculation, e.g. stripes, multi-orbital effects or nematic order, may occur on top of the physics realized here but do not seem to be the primary cause of the pseudogap observed in the cuprates via NMR probes.*

The referee points in particular to Ref. 3, which finds ‘hidden’ magnetic excitations which can be interpreted within a loop current model.

Extracting the Neutron signal of Ref. 3 is a very interesting (and challenging) task that presumably needs momentum and energy resolution beyond what we can do with the current cluster dynamical mean field technique (work based on another technique is in progress). We therefore need to defer a detailed comparison with the experimental results of Ref. 3 to a later publication. Nevertheless, we can compare our results to the theories used in that paper (Refs. 5 and 6 in that paper, here Refs. 4 and 5), which predict the establishment of loop current order in the pseudogap.

First, the single-orbital Hubbard model does not have the possibility of establishing loop current order. Despite this, we do find a pseudogap both in the single and two-particle quantities. Therefore, loop current order is not a prerequisite for a pseudogap. Second, as only short range order is present, *any* long range order is absent, excluding the mechanism of Refs. 4 and 5. We have subsumed these theories (and others of similar nature) under ‘multi-orbital effects’, as they intrinsically rely on having multiple orbitals per lattice site.

We have modified the text as follows: we have added a sentence pointing out that a comparison to the frequency and momentum-dependent susceptibility of neutron experiments remains to be done, and we have added Refs. 4 and 5 to our description of ‘multi-orbital effects’ to make it more precise. We have also found a way of adding Ref. 3 to our text, and we’ve added charge ordering as one of the phenomena that are not needed to for establishing a pseudogap. We have then

removed the statement on ‘multi-orbital effects’ from the abstract, so that it occurs together with Refs. 4 and 5 for the first time to make more clear what we mean by multi-orbital effects.

- **minor point , but in formula (5) "N" isn't defined.**
 N is the number of momentum points being sampled in the first Brillouin zone. Its definition has been added to the manuscript.
- **another minor point, it would be useful to have the units of T_1 and T_2 .**
 T_1 and T_2 are used as defined in Ref. ? .

$$\alpha T_{1\beta}^{-1}(T) = \frac{k_B T}{2\mu_B^2 \hbar^2 \omega} \sum_q^\alpha F_\beta(q) \chi''(q, \omega \rightarrow 0) \quad (1)$$

$${}^{63}T_{2G}^{-2}(T) = \left(\frac{0.69}{128}\right)^{\frac{1}{2}} ({}^{63}\gamma_n)^2 \left\{ \frac{1}{N} \sum_q F_{\text{eff}}(q)^2 [\chi'(q)]^2 - \left[\frac{1}{N} \sum_q F_{\text{eff}}(q) \chi'(q) \right]^2 \right\} \quad (2)$$

T_1 and T_{2G} have dimension $[\text{eV}]^{-1}$. We measure $(T_1 T)^{-1}$ in unit $\frac{k_B}{2\mu_B^2 \hbar^2}$, and T_{2G}^{-1} in $\left(\frac{0.69}{128}\right)^{\frac{1}{4}} {}^{63}\gamma_n$ and we have now changed the text to reflect these units.

- **I think it should be emphasized in the paper that only short range fluctuations are incorporated in the calculations. Extending the method beyond DMFT to GW would incorporate long range screening processes, which aren't accounted for by the present methodology.**
 This point is already made in the paper. In particular, the discussion section states that the method ‘treats short ranged correlations exactly and approximates longer ranged correlations in a mean field way’. In addition, we have now modified the text to say ‘These methods are controlled, in the sense that they become exact as cluster size is increased, and treat short range correlations within the cluster exactly, while longer ranged correlations are approximated in a mean field way.’ This should clarify the statement further. The combination with RPA/GW is problematic due to the quality of the GW approximation^{6,7} which does not include second order exchange terms and violates several important symmetries. We have however checked the convergence of our results by systematically enlarging our clusters, see Ref. 8.
- **I don't get the comment of the author that multi-orbital effects aren't important?**
 This comment refers to loop current order and other types of ‘orbital ordering’ in the broader sense. We believe that the answer to point 1 and the added references answers this question.
- **Maybe it would be worth mentioning what is the pseudo-gap obtained for $U=6t$, and also maybe mentioning what is the charge gap of the undoped model. Those quantities will have been published somewhere else, but reminding them in this paper would be helpful.**
 This information is already contained in the manuscript: The pseudo-gap size as a function of doping as obtained for $U = 6t$ is plotted in Fig. 1 using arrows. The charge gap of the undoped model is plotted in Fig. 2 using black open symbols.

RESPONSE TO REFEREE 2

We thank Referee B for his assessment that ‘this is a very interesting paper that should be published.’ He has ‘only a few suggestions, to clarify points for the reader’, for which we add to our revised manuscript as described below.

- lines 118-9: "Due to the divergence of lattice susceptibility near (π, π) , we use the cluster susceptibility." This confuses me -- the cluster calculation is supposed to get better as the size of the cluster increases. So how do we know that the magnetic order is really short-range, and not a long-range order that is being smeared out by the cluster averaging? Of course, the experiment says the order is only short range, but does the theory predict this? This should be carefully discussed.

Both the cluster and the lattice susceptibility are DCA approximations of the 'real' susceptibility, and both are by construction paramagnetic and caused by short ranged correlations. The distinction between 'lattice' and 'cluster' susceptibility is purely technical (see Ref. 9 and appendix for details, in short the cluster impurity is the one measured in QMC, while the 'lattice' one is a corresponding object in the thermodynamic limit). The divergence of the (π, π) lattice susceptibility can be understood as a finite size effect, and larger clusters would eventually cure this behavior. This has been investigated in detail by Maier *et al.*¹⁰, and we have added a citation to this paper to clarify.

- lines 123-4: " T_{2g} ... shows no marked change upon entering the pseudogap regime." In this instance, I don't see a clear comparison to experiment [unless that is what lines 130-1 are hinting.]

Yes, it is what line 130-1 are hinting. Fig.S8 in the supplemental material shows a comparison between experimental results and simulated results of T_{2g} . The experimental results on YBCO at different doping level shows that 1) T_{2g} keeps increasing as temperature decreases until the superconducting critical temperature T_c and 2) becomes smaller and flatter at larger doping level. Both features are seen in the simulated results. We have modified the text to reflect this.

- there are 2 typos on line 207.

Thanks for pointing that out, it has been corrected in the revised manuscript.

- The supplementary material has numerous typos, and some overlap with the main text, particularly Methods Section and figures. I believe some overlap is OK, if it is intended to maintain clarity for the reader. However, it would be good to go over this material once more, to make sure there are no unintended overlaps.

We have, as recommended by the referee, revised our supplemental material to improve readability and remove typos. The overlap in the methods section is, however, intentional and chosen to make the formulas independent and easier to read. The overlap in figures is also intentional. We aimed to simplify the side-by-side comparison with the experimental results, hence the duplication of results.

SUMMARY

In summary, we thank both referees for recommending publication and have addressed all concerns of both referees. We hope that our paper is now suitable for publication in Nature Communications.

Xi Chen
J. P. F. LeBlanc
Emanuel Gull

¹ E. Gull, M. Ferrero, O. Parcollet, A. Georges, and A. J. Millis, Phys. Rev. B **82**, 155101 (2010).

- ² H. Alloul, *Comptes Rendus Physique* **15**, 519 (2014).
- ³ Y. Li, V. Baledent, G. Yu, N. Barisic, K. Hradil, R. A. Mole, Y. Sidis, P. Steffens, X. Zhao, P. Bourges, and M. Greven, *Nature* **468**, 283 (2010).
- ⁴ C. M. Varma, *Phys. Rev. B* **55**, 14554 (1997).
- ⁵ C. M. Varma, *Phys. Rev. B* **73**, 155113 (2006).
- ⁶ J. Gukelberger, L. Huang, and P. Werner, *Phys. Rev. B* **91**, 235114 (2015).
- ⁷ H. Terletska, T. Chen, and E. Gull, *ArXiv e-prints* (2016), arXiv:1611.07861 [cond-mat.str-el].
- ⁸ X. Chen, J. P. F. LeBlanc, and E. Gull, *Phys. Rev. Lett.* **115**, 116402 (2015).
- ⁹ T. Maier, M. Jarrell, T. Pruschke, and M. H. Hettler, *Rev. Mod. Phys.* **77**, 1027 (2005).
- ¹⁰ T. A. Maier, M. Jarrell, T. C. Schulthess, P. R. C. Kent, and J. B. White, *Phys. Rev. Lett.* **95**, 237001 (2005).